# Evolution of a Cockroach Allergen into the Major Protein of Termite Royal Jelly

**DOI:** 10.3390/ijms241210311

**Published:** 2023-06-18

**Authors:** Jan A. Veenstra

**Affiliations:** INCIA UMR 5287 CNRS, Université de Bordeaux, 33600 Pessac, France; jan-adrianus.veenstra@u-bordeaux.fr

**Keywords:** salivary gland, salivary protein, major allergen, endo-β-1,4-glucanase, β-glucosidase

## Abstract

Termites live in colonies, and their members belong to different castes that each have their specific role within the termite society. In well-established colonies of higher termites, the only food the founding female, the queen, receives is saliva from workers; such queens can live for many years and produce up to 10,000 eggs per day. In higher termites, worker saliva must thus constitute a complete diet and therein resembles royal jelly produced by the hypopharyngeal glands of honeybee workers that serves as food for their queens; indeed, it might as well be called termite royal jelly. However, whereas the composition of honeybee royal jelly is well established, that of worker termite saliva in higher termites remains largely unknown. In lower termites, cellulose-digesting enzymes constitute the major proteins in worker saliva, but these enzymes are absent in higher termites. Others identified a partial protein sequence of the major saliva protein of a higher termite and identified it as a homolog of a cockroach allergen. Publicly available genome and transcriptome sequences from termites make it possible to study this protein in more detail. The gene coding the termite ortholog was duplicated, and the new paralog was preferentially expressed in the salivary gland. The amino acid sequence of the original allergen lacks the essential amino acids methionine, cysteine and tryptophan, but the salivary paralog incorporated these amino acids, thus allowing it to become more nutritionally balanced. The gene is found in both lower and higher termites, but it is in the latter that the salivary paralog gene got reamplified, facilitating an even higher expression of the allergen. This protein is not expressed in soldiers, and, like the major royal jelly proteins in honeybees, it is expressed in young but not old workers.

## 1. Introduction

In insects, IGF (insulin-like growth factor) and sirps (short IGF-related peptides) stimulate growth and reproduction [1]. This is achieved by facilitating protein production in tissues relevant for growth or reproduction, e.g., by stimulating the synthesis of vitellogenin by the fat body [2]. The worker and soldier castes of social insects neither grow nor reproduce. In termites, the soldier caste is fed by workers, as their mandibles have been modified for a defensive function that does not allow them to masticate their food [3]. They thus become dependent on workers to feed them and likely have little need for protein synthesis. Termite workers, on the other hand, consume sufficient amounts of food to stimulate protein synthesis, but they neither grow nor reproduce; instead, they feed other colony members.

Termites share food by both proctodaeal and stomodaeal trophallaxis, and, at least in some species [4,5], workers cut food into small pieces and feed those to the soldiers. In proctodaeal trophallaxis, obligatory symbionts and largely digested food are exchanged, while in stomodaeal trophallaxis, partially digested food and saliva are shared. Since protein is scarce in the wood that many termites feed on, the salivary enzymes constitute an important protein source in stomodaeal trophallaxis. In some higher termites, stomodaeal trophallaxis exclusively concerns salivary gland secretions [6,7] and thus resembles the production of royal jelly in honeybees that is fed by bee nurses to the queen, young larvae and those older larvae that are allowed to develop into reproductives.

An analysis of various publicly available termite genomes and transcriptomes showed that termites have one IGF and four different sirps, which have been called atirpin, brovirpin, cirpin and birpin, respectively. The expression of the four different sirps in different termite species suggested that each of them may have a specific function [8]. Brovirpin is strongly expressed in the ovary, suggesting an important role in vitellogenesis, while birpin seems to be important for stimulating growth in larvae destined to become soldiers or reproductives, and atirpin is likely to be a general tissue factor. In workers, the expression of both birpin and brovirpin is weak, which would be consistent with the absence of either growth or reproduction in this caste. Yet, there must be protein synthesis in the salivary glands of larvae, the major source of protein for larvae and queens in termite colonies. In higher termites, worker saliva is the only food for physogastric queens and functions like royal jelly in honeybees; in other words, it becomes termite royal jelly. Therefore, one might perhaps expect IGF or one of the sirps to stimulate protein synthesis in the salivary glands. In order to test this speculative hypothesis [8], it would be useful to know the coding sequences of the proteins produced by the salivary glands in higher termites.

Whereas the saliva from soldiers has been studied in some detail ([9] and references therein), and the major proteins of honeybee jelly are well known [10], little is known of the saliva from higher termite workers. Cellulose-digesting enzymes have attracted a lot of attention, and several termite cellulose-digesting enzymes have been characterized and found to be expressed predominantly in the salivary glands of lower termites. The endogenous termite enzymes consist of a number of closely related endo-β-1,4-glucanases from the glycoside hydrolase family 9 (GHF9) and a single β-glucosidase [11,12,13,14,15,16,17]. In the lower termites, the SDS protein gel analysis of termite salivary glands revealed a major protein of about 49 kDa in *Prorhinotermes simplex* (Hagen, 1858), *Reticulitermes flavipes* (Kollar, 1837), *Coptotermes formosanus* (Shiraki, 1909), *Kalotermes flavicollis* (Fabricius, 1793) and *Zootermopsis angusticollis* (Hagen, 1858) [9]. A partial protein sequence of this protein was obtained from *P. simplex* and allowed it to be identified [9] as an endo-β-1,4-glucanase previously characterized from other termites (e.g., [13,15,16]). In the higher termites, endo-β-1,4-glucanase is not expressed in the salivary gland but rather exclusively in the midgut [18,19]. In two species of the Termitidae, a much larger salivary protein of about 85 kDa was observed on SDS protein gels, while 49 kDa proteins were not detected. The N-terminal sequence of the 85 kDa protein from *Neocapritermes taracua* (Araujo & Krishna, 1968) revealed it to be related to a major cockroach allergen [9] known from cockroaches [20]. These proteins have been studied extensively in *Blattella germanica* (Linnaeus, 1767) and *Periplaneta americana* (Linnaeus, 1758), as they are major allergens (MAs) that can induce asthma. They are produced in the midgut and persist in cockroach feces, the source of the allergens in infested habitats when these are not thoroughly cleaned, but their exact function in digestion is unknown (e.g., [20,21,22]). Lysozymes have also been reported from salivary glands in worker termites, but these do not appear to be expressed in very high levels [19,23].

The availability of five termite genome assemblies [24,25,26,27,28] and a significant number of termite transcriptome SRAs provides the opportunity to explore these salivary proteins in more detail.

## 2. Results

### 2.1. Cellulose-Digesting Enzymes

Termites typically have a number of β-1,4-endoglucanases genes, most of which have very similar nucleotide sequences coding for enzymes that vary in only a small number of amino acid residues. They also have one gene that, although belonging to the CHF9 group, is distinctly different. I will refer to the first group as GHF9-1 enzymes and the other one as CHF9-2. The latter is an ortholog of a phasmid protein from *Timema cristinae* (Vickery, 1993) that has no detectable β-1,4-endoglucanase activity [29] and for which orthologs have been described from other Polyneoptera that similarly lack β-1,4-endoglucanase activity [30]. When only next-generation sequencing technology with short sequences is available to assemble a genome, it is difficult to complete the GHF9-1 genes, as their coding sequences are so similar. It is for this reason that the genes coding the GHF9-1 enzymes are often poorly represented and possibly incorrectly reconstructed in termite genome assemblies. Transcriptome assemblies obtained from short-read sequences have the same problem, and in this case, it can be worse due to the presence of single-nucleotide polymorphisms. The *Zootermopsis nevadensis* genome assembly has nine different contigs that code for significant pieces of GHF9-1 enzymes, but several are incomplete, and it is plausible that there are only five or six of such genes. One of the two genome assemblies of *Reticulitermes speratus* (Kolbe, 1885)—the unpublished one (https://www.ncbi.nlm.nih.gov/Traces/wgs/?val=JAJKFL01, accessed on 3 March 2023)—lacks all evidence for GHF9-1 genes, even though transcriptomes unequivocally reveal their existence; the second *R. speratus* genome assembly [28] has one contig that contains two and a half GHF9-1 genes and three contigs that contain partial GHF9-1 genes. In the current *Coptotermes formosanus* genome assembly, there are two complete β-1,4-endoglucanase genes. One of these is, with the exception of a single nucleotide, identical to one of the two β-1,4-endoglucanases previously identified and characterized from this species from expression libraries [15,16]. The second one is different, meaning that there must be at least three GHF9-1 enzymes in this species. The *Cryptotermes secundus* (Hill, 1925) genome assembly has three and a half GHF9-1 genes, of which the partial one may be an allele of one of the other three. The *Macrotermes natalensis* (Haviland, 1898) genome assembly contains evidence for four such genes, but only one GHF9-1 gene looks complete. The number of GHF9-1 genes was estimated by determining the number of spots in genome SRAs coding the exon containing the signal peptide of these genes. The results suggest five such genes in *Zootermopsis*, possibly more than six in *Reticulitermes*, between three and five in *Cryptotermes* and probably about 5 or 6 in *Macrotermes*. Raw sequence reads for constructing the *Coptotermes* genome are not publicly available, and, therefore, these cannot be used to estimate the number GHF9-1 genes in this species.

On the contrary, the single GHF9-2 enzyme does not have this problem, as its primary amino acid and coding sequences are significantly different from the GHF9-1 genes. Consequently, it appears correctly assembled in the various termite genomes.

### 2.2. Major Allergens

Although their primary amino acid sequences are variable, similar proteins can be found in many insect species (Figure 1). Their predicted MA amino acid sequences start with a signal peptide followed by a variable number of peptide repeats of around 190 amino acid residues, which look like the active principle coded by these genes and which I refer to as the repeat sequence. Interestingly, these repeat sequences are themselves tandem repeats of two sequences of around 100 amino acid residues [20]. The number of repeat sequences varies from one to six within a single MA gene. Within each gene, the repeat sequences are very similar to one another. In termites, these genes have one exon coding the signal peptide and three exons coding for each repeat sequence (Figure 2).

The MA gene is commonly amplified in insects. In the *Zootermopsis nevadensis* genome assembly, there is only one MA gene coding for a protein that has two repeat sequences, while in *Bombyx mori* (Linnaeus, 1758), there are two paralog MA genes that each have one repeat sequence. Whereas the *Zootermopsis* and *Bombyx* genomes were produced from large individual sequence reads, the *Periplaneta americana* and *Blattella germanica* genomes were sequenced using short illumina sequences. As the raw sequences used for these cockroach genome assemblies are small and the cockroach MA genes are more numerous and contain more repeated sequences, their assembly is problematic. For example, in the *Periplaneta* genome SRA SRR16925037, there are 127 individual reads that code for the signal peptide of its MA, while the expected number for a single-copy gene is 30. This suggests that there are about four paralog *Periplaneta* MA genes, while the database shows several different *Periplaneta* MA transcripts. The abundance of MA genes combined with the repeated sequences makes the assembly of transcripts and genes very difficult. The third *Periplaneta* genome assembly, obtained by a group working on cockroach allergens, still did not allow for their unambiguous assembly [31].

Using one of the *Periplaneta* MA sequences as a query for the blastn_vdb command shows it to be expressed predominantly in the midgut (Appendix A). In other insect species, they are similarly expressed in the midgut, but in honeybees they are also expressed in the Malpighian tubules and Nasonov and venom glands, albeit at much lower levels (Appendix A). The identification of the tissues expressing one such gene in *Blattella* has been studied using both RT-PCR and specific antibodies. Both techniques reveal the midgut as the predominant tissue expressing the gene [21,32,33]. Based on the number of spots for these proteins in midgut SRAs, these genes are sometimes fairly highly expressed; their reads may make up from 0.5 to 1.5% of the total number of reads in midgut SRAs (Appendix A). When mosquitoes are fed sugar water, the MA expression in the midgut is virtually zero, but it becomes high after blood feeding [34]. Similarly, in the German cockroach, expression is much lower during pregnancy, when females feed less [21]. This suggests a role in digestion or perhaps more generally in protein secretion, since MA is also expressed in the honeybee Nasonov end venom glands.

As mentioned, *Zootermopsis* has a single MA gene that codes for a protein with two repeat sequences. *Reticulitermes*, similarly, has one such gene, but its MA has three repeat sequences, while *Cryptotermes* also has a single MA gene that appears to code for a protein with five repeat sequences. The raw sequence data used for the *Coptotermes* genome assembly are not publicly available, and it is therefore not completely clear how many MA genes this species has. The published genome assembly shows one MA gene with a single repeat sequence. Transcripts (GCET01093399.1, GCET01105335.1 and GHZJ01147378.1) from other *Coptotermes* species suggest that this might be incorrect.

The *Macrotermes* genome assembly [25] has a number of contigs containing MA coding sequences. One of these contains a sequence that seems to be present as a single-copy gene coding an MA with two repeat sequences, I called this gene MA-1. The other contigs contain sequences that are mostly incomplete and code for a different type of MA, which I called MA-2. Two scaffolds each contain parts of two such genes. Using the exon for the signal peptide of MA-1, as a query in a tblastn search on one of the raw genomic reads (SRRSRR789336) from this species, only four individual reads are found, whereas the average number of reads for a coding sequence of the same size of eleven single-copy neuropeptide genes was 6.6 (range 2 to 12), confirming that this gene most likely exists as a single copy in the genome. Two very similar signal peptide coding exons from the other MA-2 coding contigs, which are the same length as their MA-1 homolog, yielded a total of 28 reads, suggesting about four MA-2 genes. Reads coding the second coding exons of MA-2 yield about 20 times the number expected for a single-copy gene with one repeat. If MA-2 paralogs have four repeats each (this might correspond to the observed molecular weight of these proteins on SDS gels [9]), it would suggest about five paralog MA-2 genes.

Even though the exact sequences of the various *Macrotermes* MA-2 proteins are not known, the amino acid compositions are strikingly different from those of the MA-1 proteins from this and other termite species. Thus, whereas the amino acids cysteine, methionine and tryptophan are lacking from MA-1, they are present in several copies in MA-2 (Figure 3). These differences in the amino acid composition between MA-1 and MA-2 are striking. Although a few MA-1 proteins have one or two of those three amino acid residues, the MA-2 proteins always have at least twelve, as far as this can be ascertained from partial transcripts. The only exception may be a partial transcript from a *Hospitalitermes* species that looks like it might be an intermediary. MA-2 protein transcripts were found in some but not all species (Appendix A). The genome assemblies from *Zootermopsis*, *Reticulitermes* and *Cryptotermes* each show only one MA gene, all of the MA-1 type, while the analysis of the genome SRAs of these species fails to reveal any evidence for a MA-2 gene. The salivary gland transcriptome SRA from *Coptotermes formosanus* revealed large numbers of MA-1 reads but no evidence for MA-2, and the genome assembly of this species does not show any evidence for an MA-2 gene.

MA-2 genes are absent from the genome assemblies of four lower termites. Although this might suggest that MA-2 is specific for the higher termites, this is not so. Partial transcripts for other MA-2s, i.e., MAs that are relatively enriched in methionine, cysteine and tryptophan, can be identified in transcriptomes from lower termite families, such as the Rhinotermitidae, Kalotermitidae and Stylotermitidae. Thus, partial transcripts (Appendix A) confirm the existence of an MA-2 gene in *Porotermes quadricollis* (Rambur, 1842), *Kalotermes flavicollis*, *C. secundus* and *Heterotermes tenuis* (Hagen, 1858).

The absence of essential amino acids in termite MA-1 is not surprising, as essential amino acids are particularly scarce in termites. At first sight, it is thus surprising that a paralog incorporates several of these essential amino acid residues, when it appears that MA-1 without these amino acid residues is likely to be fully functional. However, in higher termites, the only food queens receive is the saliva secreted by workers. In *N. taracua*, the major saliva protein was identified as a major allergen. Of the two types of major allergens coded by termite genomes, only MA-2 would be able to provide for all essential amino acids needed. There can thus be no doubt that MA-2 in higher termites is expressed in the salivary gland and essentially functions as a food source for the queen and other colony members. Although there is no proof that this is the only tissue expressing MA-2, this seems very likely. Indeed, it is tempting to suggest that those lower termite species that have MA-2 similarly express it in the salivary gland.

The presence of MA-2 transcripts in several lower termite families shows that such proteins originated relatively early during termite evolution, while a sequence similarity tree of the termite MAs suggests that the various MA-2s may have evolved from a common precursor (Figure 4). Nevertheless, MA-2 is absent from several species, and it cannot be excluded that MA-2 evolved repeatedly.

### 2.3. Lysozymes

Lysozymes have also been identified in salivary glands from worker termites. Termites have a large number of genes coding for lysozymes; however, as described below, only two of those show significant expression in the salivary gland from *C. formosanus*.

### 2.4. Expression

The only salivary gland-specific SRA is from a lower termite species, *Coptotermes formosanus.* This SRA contains numerous spots for β-1,4-endoglucanases as well as a significant number of spots corresponding to MA-1, β-glucosidase and two lysozymes. Of the various termite lysozymes that can be identified from its genome, these are the only two that are prominently expressed in the salivary gland (Appendix A).

In experiments with *Reticulitermes labralis* (Hsia & Fan, 1965), forty workers were isolated together with two soldiers [35]. In the absence of larvae, the expression of the β-1,4-endoglucanase gene decreased significantly. In transcriptome SRAs from workers, β-1,4-endoglucanase spots make up, on average, eight percent of all reads. Once workers are isolated with a few soldiers but no longer have to care for larvae, this percentage falls to three percent, while the synthesis of storage proteins increased significantly in the same insects. Under these conditions, a few isolated workers develop into neotenic reproductives, and the percentage of β-1,4-endoglucanase genes in these neotenic reproductives is even lower (Figure 5). Although there are insufficient data for a statistical analysis, the differences observed are sufficiently large to suggest their physiological relevance.

Although it is obvious that MA-2s are expressed by a large number of species and perhaps by every higher termite, there are very little SRA data allowing us to look at the expression of MA-2 under different conditions or in different castes. Useful data are only available for two *Macrotermes* species, and even those data are rather limited. A recent study compared the molecular underpinnings of the division of labor in minor workers of *M. bellicosus* (Smeathman, 1781). The authors produced several transcriptome SRAs from heads and thoraces, which thus include the salivary gland [36]. An analysis of these SRAs shows that the expression of MA-2 is about twice as high in the builders as it is in the foragers (Figure 6). In an earlier study, the same group used the same species, but in that case, only the heads were analyzed [37]. Those results show a larger number of MA-2 reads in younger workers than in older workers in both minor and major workers, while young major worker have a larger number of MA-2 spots than young minor workers (Appendix A). However, as these SRAs are from heads [37], they should not contain the salivary glands, or perhaps only a small part of it. Since the number of samples is also very small, it is unclear how reliable these differences really are.

There is one SRA dataset from *M. natalensis* [38] that reveals a high expression of MA-2 in whole bodies from workers and a very low expression in queens and kings, except for one young queen (Appendix A).

## 3. Discussion

This manuscript deals with proteins that are highly expressed in the termite salivary gland. In termites, both MAs and β-1,4-endoglucanases are often coded by paralog genes with remarkably similar primary amino acid sequences. Gene duplication is a common phenomenon and often leads to the paralogs evolving into two distinct proteins, each with its own function. In most cases, this type of gene duplication is easily recognized in genome assemblies, as the amino acid sequences of proteins coded by the paralogs usually diverge over time. However, when the advantage of the duplication of the gene is, at least initially, the concomitant increase in protein production, protein and coding sequences may remain very similar and diverge little. This type of gene duplication can cause significant problems in the assembly of their genes and transcripts, even more so when the assembly is using exclusively short-read technology.

β-1,4-endoglucanases are the major proteins detected in the salivary gland of several lower termite species [9]. Genome and transcriptome data reveal that there are a number of genes that code for proteins with similar primary amino acid sequences. These sequences are very well conserved, suggesting they have the same or very similar functions. Indeed, a similar multitude of β-1,4-endoglucanase GHF9-1 genes is expressed in the midgut of phasmids. Impressive work in one of these species has nicely illustrated that they are all β-1,4-endoglucanases but have slightly different substrate specificities. The clustering of some of the termite GHF9-1 genes on the same contigs suggests that they originated from successive local gene duplications, not unlike amylase genes in animals that consume a significant amount of starch. The single termite GHF9-2 enzyme has orthologs in other Polyneoptera, and these lack β-1,4-endoglucanase activity in at least some species [29,30].

One set of data indicates that termite salivary β-1,4-endoglucanases do not merely function as digestive enzymes but also constitute a significant food source for larvae. Thus, in *R. labralis*, the expression of the β-1,4-endoglucanase genes decreased when they no longer had to feed larvae. Yet, there is no evidence that these animals are starving as a result of less salivary β-1,4-endoglucanase. On the contrary, the synthesis of storage proteins is significantly increased, indicating that they are well fed. Indeed, a small number of such workers will develop into secondary reproductives [35], and such animals even further decrease the production of β-1,4-endoglucanase. This suggests that a significant amount of β-1,4-endoglucanase transferred to larvae by stomodaeal trophollaxis does not contribute to the digestion of food by the workers themselves but rather serves as a protein source for other members of the colony. Once the primary reproductives in an *R. speratus* colony no longer have to feed the larvae because there are sufficient workers to do so, the expression of β-1,4-endoglucanase similarly decreases [39].

When saliva is shared as food by trophallaxis, it constitutes an important source of protein. However, a priori, it is unlikely that their amino acid composition is optimal as a food source to ensure larval growth. Fortuitous mutations in the amino acid sequence of a β-1,4-endoglucanase that might improve its amino acid composition must be expected to interfere with their enzymatic activity, since the primary amino acid sequence of these enzymes is so strongly conserved. Other salivary proteins might compensate for the less-than-ideal amino acid composition of these proteins more easily. Three other types of proteins have been described from termite salivary glands, β-glucosidases, MAs and lysozymes. Although the amino acid sequences of lysozymes appear more variable, this may reflect the existence of different types of lysozymes, while the amino acid sequence of β-glucosidase is also critical for its enzymatic activity.

The larger amino acid sequence variability of MAs suggests that their function—whatever that may be—will more easily tolerate fortuitous mutations that would improve their nutritious value as a protein source. Natural selection may thus be expected to favor mutations in MAs rather than in β-1,4-endoglucanases, β-glucosidase or lysozymes to increase the nutritious value of saliva. This development may have been facilitated by the amplification of the termite MA gene, an increase in the number of repeats in each MA as well as the preferential expression of the MA-2 paralog in the salivary gland. The functions of the MA-1 and MA-2 paralogs then diverged and led to the inclusion of methionine, cysteine and tryptophan residues in MA-2. The presence of MA-2 genes in many lower termites suggests that this happened relatively early during evolution, perhaps independently on separate occasions. Once higher termites no longer needed salivary β-1,4-endoglucanase for cellulose digestion, the amplification of the MA-2 gene allowed for its expression in much larger quantities. At the same time, due to there being multiple copies of the MA-2 gene, mutations that further optimized the amino acid composition may have accumulated more rapidly. The presence of a single large protein band on SDS gels of salivary glands does not exclude the possibility that this protein band represents a collection of several closely related MA-2 proteins produced from different paralogous genes that have very similar sizes and amino acid sequences.

There are interesting similarities between the major royal yolk proteins from honeybees and the MA-2s from the Termitidae. Honeybee royal jelly contains a mixture of highly homologous major royal jelly proteins (MJRPs). Nine genes coding MRJPs are located next to one another on the same chromosome fragment, flanked on both sides by yellow proteins, from which they evolved by local gene duplications [10,40]. The *Macrotermes* genome assembly lacks sufficiently long contigs to confirm that the various MA-2 copies are similarly located on the same chromosomal fragment, but this is probably the case, since some scaffolds contain parts of more than one MA-2 gene. Furthermore, the various MAs are highly homologous and have the same intron splice sites, which is also consistent with local gene duplications. In honeybees, the MJRPs are expressed in young worker larvae but not in males, queens or older larvae. This is similar to the expression of MA-2 in *Macrotermes* termites.

There is also a significant difference. When new honeybee colonies are started by freshly emerged queens, they take large numbers of workers with them to establish a new colony. Termites, on the other hand, start a colony with only the royal pair, and the founding queen and king are responsible for feeding the first emerging larvae (e.g., [6]). It is interesting to see that, in a transcriptome SRA of one of the younger *M. natalensis* queens, there are large numbers of spots corresponding to MA-2 transcripts, suggesting that this queen still participated in feeding her descendants. One might expect to also see this in younger kings [38]. Workers and reproductives of incipient colonies might be good subjects for studying the possible implication of IGF and/or sirps in the stimulation of protein production by the salivary glands. Currently, there are insufficient publicly available data to do so.

It is tempting to speculate that the expression of MA-2 in higher termites might be a useful marker for their behavioral role in a colony. Preliminary data on the difference in the expression of MA-2 in *M. bellicosus* between foraging and building minor workers are promising in this respect. It will be interesting to see whether or not the differences observed in MA-2 expression between minor and major workers of the same species are genuine. If they are, it might indicate that the major workers have a more important role in providing sustenance for other members of the colony. It seems reasonable to assume that a large physogastric queen producing more than 10,000 eggs per day needs to be fed at a high speed. Major workers are larger and can thus provide larger meals, but if they indeed were also to produce relatively more protein, they might be the preferred providers for such queens.

## 4. Materials and Methods

Genome assemblies for five termite and two other cockroach species are publicly available and were downloaded from NCBI (https://www.ncbi.nlm.nih.gov/genome/?term=blattodea, accessed on 3 March 2023) and, in the case of the *Macrotermes natalensis* genome, from GigaDB (http://gigadb.org/dataset/100057, accessed on 3 March 2023). They were analyzed using Artemis [41] and the BLAST+ program (https://hpc.nih.gov/apps/Blast.html, accessed on 3 March 2023). The termite transcriptome and genome short-read archives (SRAs) were downloaded from NCBI and analyzed using the SRA toolkit (https://hpc.nih.gov/apps/sratoolkit.html, accessed on 3 March 2023) and Trinity [42], following the methods described in detail elsewhere (e.g., [8,43]). When available, termite transcriptomes at NCBI were also used. Obviously, these transcriptomes were not produced to answer the question of which proteins might be expressed by the salivary glands in workers. Indeed, there is only a single termite worker salivary gland SRA publicly available, from *Coptotermes formosanus*, a lower termite species.

This manuscript deals with three different types of proteins, cellulose-digesting enzymes, i.e., endo-β-1,4-glucanases and β-glucosidase, proteins that were initially identified as major cockroach allergens (MAs) and, more succinctly, lysozymes. With the exception of β-glucosidase, the other proteins often have very similar paralogs, which makes it difficult to correctly assemble coding sequences using short illumina reads from genome or transcriptome SRAs. Furthermore, MAs have a variable number of repeat sequences, which further increases the difficulty of sequence assembly. This problem is particularly significant in *Macrotermes natalensis*, as there may be five different MA genes, four of which likely have four repeat sequences each.

In order to estimate the number of genes coding these proteins, I determined the number of spots coding their signal peptides in genome SRAs and compared those numbers with the average number of reads for coding sequences of an identical length of single-copy genes. Signal peptides tend to be better conserved than the rest of a protein sequence, as their only function is to ensure the transfer to the endoplasmic reticulum, and this function remains the same in a newly evolved paralog protein. The identity of such coding exons is further confirmed by the presence of a conserved intron donor site.

Phylogenetic trees consist of diagrams joining different species based on the sequence similarity of one or more proteins or genes. When the data yield branching patterns that are statistically fixed, such trees represent the phylogenetic relations between the species involved. However, in many instances, the probabilities of specific patterns are less than one, and in such cases, it seems incorrect to call such diagrams phylogenetic trees. The tree made for the different termite MA sequences was based on the signal sequence and the first tandem repeat sequence of these proteins. These sequences are derived from genome assemblies and—often partial—transcripts from publicly available transcriptomes at NCBI (https://www.ncbi.nlm.nih.gov/Traces/wgs/?term=tsa, accessed on 3 March 2023).

To analyze protein expression, the relative number of spots corresponding to their coding sequences was determined in transcriptome SRAs. The number of such spots was determined using the blastn_vdb command from the sratoolkit, as described in detail elsewhere [8,44]. Insect salivary glands are typically present in the thorax, and this is also the case in termites [7,45]. Many termite transcriptome SRAs are made from the head and thorax and can thus be used to look at the expression of putative salivary proteins. However, others are made from the head only and probably contain no or very little tissue from the salivary gland.

When it is impossible to obtain complete coding sequences for a protein, as is the case here, e.g., for some of the allergen paralogs from *M. natalensis* and *M. bellicosus*, the question arises as to whether this method is reliable. When counting the number of spots for MA-1 in the various *M. bellicosus* SRAs using the coding sequence of *M. natalensis* MA-1, one obtains, on average, 93.5% of the counts obtained when using the incomplete coding sequence of *M. bellicosus* MA-1. For MA-2, the corresponding number is 90.5%. Although these numbers are not perfect, they are good enough to establish the relative expression levels of these proteins. The sequence similarity between MA-1 and MA-2 is such that some spots are identified in both the MA-1 and MA-2 queries. This number was less than 0.5% of the total number of reads.

Comparing gene expression using SRAs is complicated, as each SRA has an n = 1. Given the small numbers of samples, it is not realistic to test whether samples follow a normal distribution. Thus, only parameter-free statistics can be used, and comparing different castes introduces additional restraints on the analysis (see [8] for a more detailed discussion). A two-sided Wilcoxon rank sum test was used in the sole statistical test applied here.

## 5. Conclusions

In lower termites, the salivary proteins that are fed to larvae and reproductives appear to be mostly cellulose-digesting enzymes that lack certain essential amino acids. The gene coding a paralog of a major cockroach allergen has been amplified, and these newly evolved genes code for proteins that incorporate these essential amino acids and seem to be the major proteins present in the saliva of workers in higher termites. The amplification of a preexisting gene, its repurposing into a nutritious protein and its subsequent reamplification and strong expression by exocrine glands resemble the major proteins in honeybee royal jelly.

## Figures and Tables

**Figure 1 ijms-24-10311-f001:**
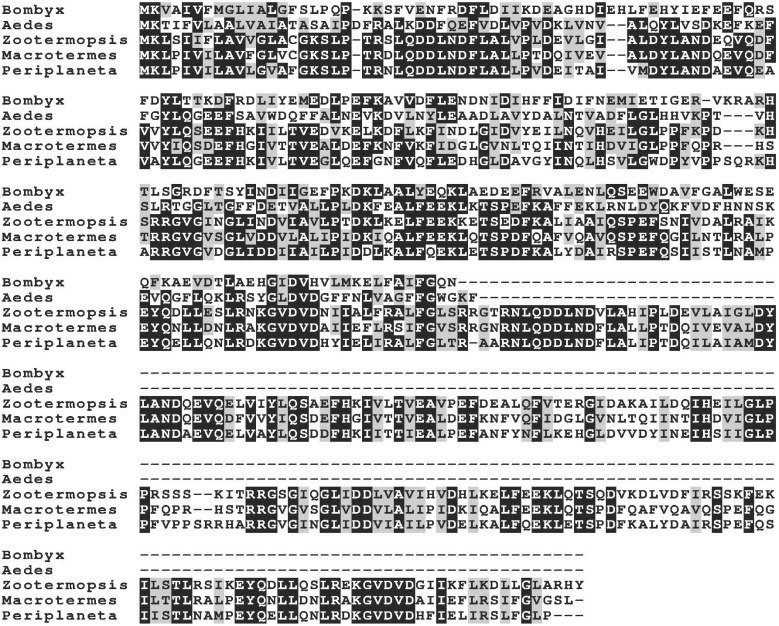
Sequence alignment of MAs from five species: the mosquito *Aedes aegypti* (XP_001660827.1), the silk worm *Bombyx mori* (corrected from XP_037876368.1), the cockroach *Periplaneta americana* (GEIF01009400.1) and the termites *Zootermopsis nevadensis* (XP_021939355.1) and *Macrotermes natalensis* (MA-1, identified here from the genome assembly). The termite sequences shown here have two repeat sequences, while the *Aedes* and *Bombyx* MAs have only one. Note that these proteins show significant sequence variability.

**Figure 2 ijms-24-10311-f002:**
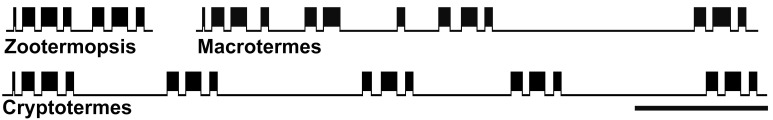
The exon-intron structures of three MA genes from the termites *Zootermopsis nevadensis*, *Macrotermes natalensis* and *Cryptotermes secundus*. The thin black line indicates the untranslated DNA sequences, and the thick parts indicate the exons. Note that the untranslated 5′ and 3′ ends have not been indicated, as they could not be reliably determined. These genes consist of a small exon for the signal peptide followed by one or more assemblies of three exons that each code for one repeat sequence. The *Zootermopsis* and *Cryptotermes* are MA-1 genes, and the *Macrotermes* is an MA-2 gene. The figures have been drawn to the same scale (the scale bar corresponds to 2000 base pairs).

**Figure 3 ijms-24-10311-f003:**
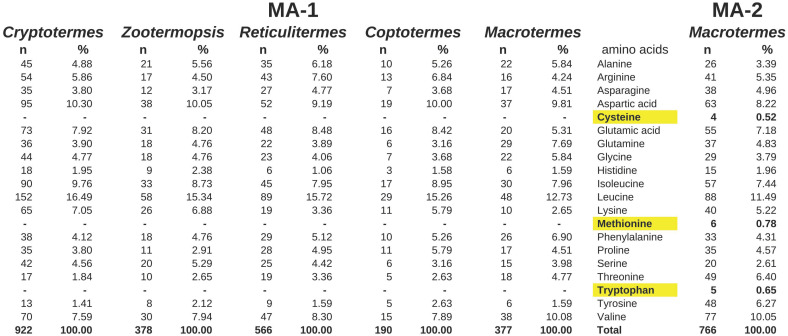
Amino acid composition of the predicted MA-1 proteins from five termite species and one MA-2 from *Macrotermes natalensis*. Note that three essential amino acids, highlighted in yellow, are completely lacking from all the MA-1 proteins, whereas they are incorporated in the MA-2 protein. All termite sequences here were either deduced from the genome assemblies of these species. *n*, number of residues of the amino acid present in the protein; %, percentage of the amino acid of the total.

**Figure 4 ijms-24-10311-f004:**
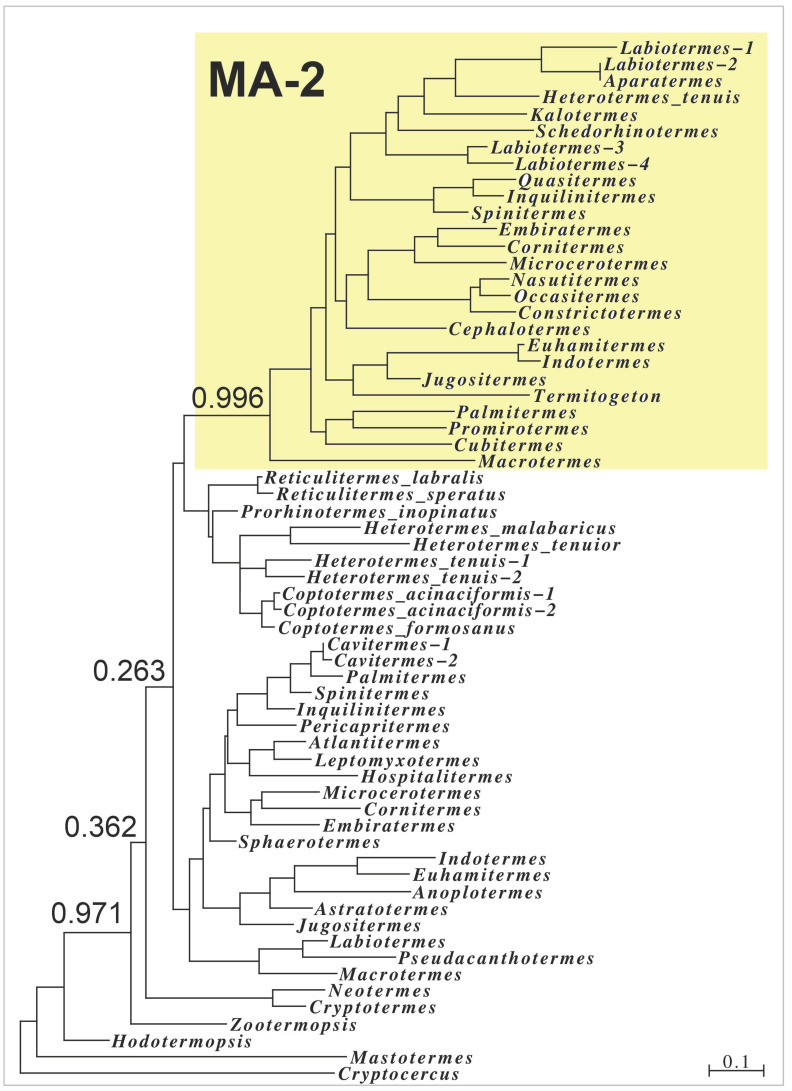
Sequence similarity tree of termite MA sequences based on the sequences of the signal peptide and the first repeat sequence. Note that the MA-2 sequences, highlighted in yellow, form a very distinct branch on the tree with high probability; all other sequences are MA-1 proteins and do not contain significant amounts of methionine, cysteine nor tryptophan residues.

**Figure 5 ijms-24-10311-f005:**
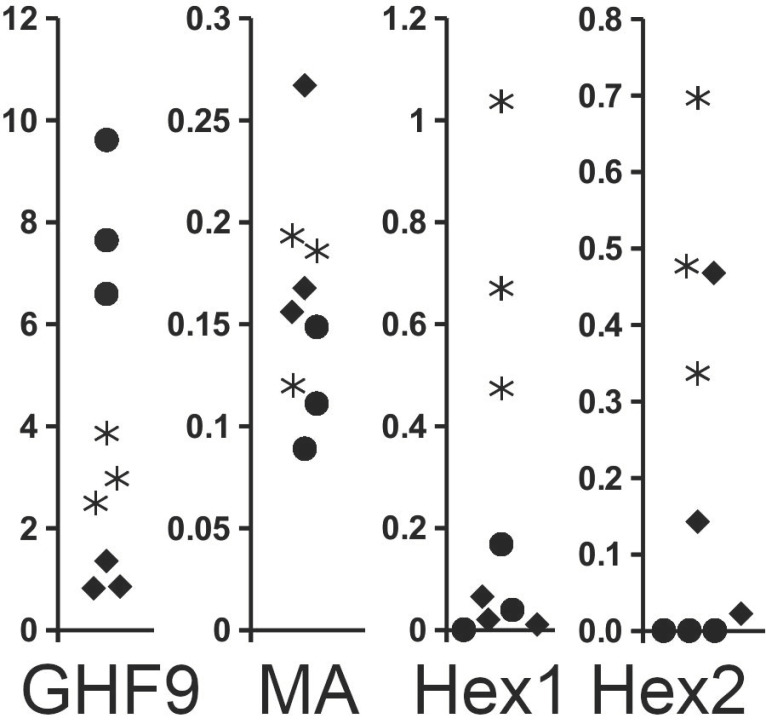
Changes in the expression of four different proteins, as determined from SRA transcriptome data analysis in workers from *Reticulitermes labralis* that have been isolated from the colony in groups of forty workers and two soldiers. Data are expressed as the percentage of the number of spots in an SRA for each protein. Note that the expression of β-1,4-endoglucanase diminishes rapidly once the workers are isolated from the colony, while at the same time, the expression of both hexamerins increases significantly. Closed circles: workers from a normal colony; asterisks: isolated workers; diamonds: neotenics that developed from isolated workers. GHF1, β-1,4-endoglucanase; MA, major allergen; Hex1, tyrosine-rich hexamerin; Hex2, methionine-rich hexamerin. Data extracted from SRAs from reference [35].

**Figure 6 ijms-24-10311-f006:**
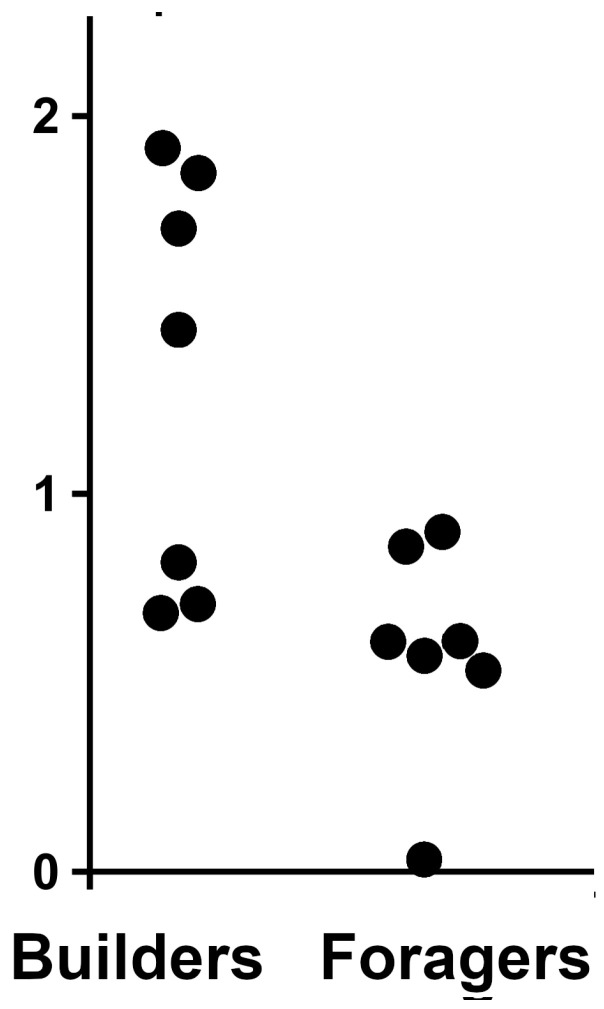
Difference in the expression of MA-2 in minor workers of *Macrotermes bellicosus* that are engaged in either building or foraging. Values are the percentage of MA-2 spots in SRAs. Original SRA data from reference [36]. *p* < 0.02.

## Data Availability

Individual SRAs used here are all listed in the Appendix A.

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
