# Peer review of "Evolution of a Cockroach Allergen into the Major Protein of Termite Royal Jelly"

_ijms, 2023, doi:10.3390/ijms241210311_

Round 1
Reviewer 1 Report
General comments
This is an interesting manuscript bringing light on the composition of termite saliva in higher termites. The manuscript is well written, presented and discussed. The manuscript was based on genome assemblies publicly available and downloaded from NCBI. The genomes were analyzed and compared among five termite and two other cockroach species. The manuscript is in the scope of the journal.
Specific comments
Key words: I suggest including other terms different from those ones used the title.
Introduction: According to the international rules of zoological nomenclature, the first time the species name is cited it must be followed the name of its authors description. Please check the scientific names not just in the introduction but in the entire manuscript. Also, “termitidae” is a Family name and must be written as “Termitidae”
Discussion
Lines 379- 389 Several references are missing in this paragraph.
I would suggest a separated topic for Conclusion
Line 381- Please check “ It are the founding” is it correct? This could be an English language mistake? If so please check the entire manuscript
I am not a expert in English language. I had no dificulties to understand the text. Please check the comments bellow.
Author Response
Thank you for your time in reviewing my manuscript. I have addressed all of your concerns. I have added the authors who first described the species when they are mentioned for the first time in the manuscript. I have corrected the various typo's, I have added a conclusion and I have also added a few references where they were missing.
Your final question whether “ It are the founding” is correct English surprised me somewhat, as it seemed correct to me. Some information on the internet
https://ell.stackexchange.com/questions/11305/it-is-or-it-are
suggests that indeed it should be correct, but to avoid confusion I changed the sentence, also because a second reviewer did not like it either.
Reviewer 2 Report
General comments
Title is somewhat deceptive as the author argues in a circular way that what he is describing is a royal jelly, although the exact composition is not proved. An assumption is present in the general introduction regarding because the larvae and queen are fed from the salivary gland material, there must be protein synthesis in those glands. Suppose the termites are more like honey bees and the “jelly” is produced in an alternate “undescribed” or “unrecognised” gland? Maybe better to change the title to a question as that is more the way the work is presented as result of few instances of only labial gland sequences that are required to ensure that the gene products are not represented by other components of the body.
Argument regarding reduction of expression of ß-1,4-endoglucanase relies on that expression decrease leads to a decrease in enzyme. Without measuring the activity of the enzyme in isolated glands, this is highly subjective. This maybe better as a suggestion for further directed work rather than support for the stated hypothesis.
The “common sense” rule for statistics is not necessarily appropriate, as “common sense” is determined by the author to support their argument. Maybe best to present the data graphs and state that further work is necessary to corroborate the argument presented.
Minor comments
Line 96. Should “If” be “I”?
Line 123. Change “theses” to “these”
Line 134. Reword “variable number of a peptide of around”
Line 149. “he” should be “the”
Line 161. Should “illumina” be capitalised?
Line 187. Delete second “that”
Line 191. Delete “Whereas”
Line 211. Change “methione” to “methionine”
Line 255. Change “ß-glocosidase” to “ß-glucosidase”
Line 258. Change “fourty” to “forty”
Line 304. Change “involving” to “evolving”
Line 381. “It are” needs rewording
Line 383. Change “an transcriptome” to “a transcriptome”
Line 431. Change “consists” to “consist”
Line 513. “Blatella germanica” should be italicised.
No comments other than minor corrections in review
Author Response
I thank you for the time you spent on reviewing my manuscript. You had several comments and I will try to address them all here:
Your first comment:
Title is somewhat deceptive as the author argues in a circular way that what he is describing is a royal jelly, although the exact composition is not proved.
The manuscript deals exclusively with the proteins in termite saliva, not with any other components. So I don't understand why the title suggests that I would describe the complete composition.
Your second comment:
Title is somewhat deceptive as the author argues in a circular way that what he is describing is a royal jelly, although the exact composition is not proved. An assumption is present in the general introduction regarding because the larvae and queen are fed from the salivary gland material, there must be protein synthesis in those glands. Suppose the termites are more like honey bees and the “jelly” is produced in an alternate “undescribed” or “unrecognised” gland?
It may look a reasonable hypothesis at first sight that termite workers would have an “undescribed” or “unrecognized” gland that is responsible for the secretion of the secretions fed to the royal pair and larvae. When I first started to look at this I had the same question. However, such a gland would be of a certain size (the honeybee hypopharyngeal gland is not small either) and it would have been discovered by now as the salivary glands of termites have been extensively researched. Also, several termite researchers call this the saliva, implying it comes from the salivary gland. In lower termites the salivary gland produces large amounts of cellulose digesting enzymes and those are transferred together with partially digested wood to other members of the termite colony. In lower termites expression of these cellulases have been demonstrated by both enzymatic assay and RT-PCR to be localized to the salivary gland. Certainly, if there were yet another gland that made these enzymes, it would have been detected. In higher termites a partial protein sequence of MA was determined. As I describe here termites have two types of MA, one that lacks essential amino acids and thus can not serve as a complete protein source, and second one that does contain these essential amino acids and hence can serve as a complete protein source. The genes for this second type of MA is amplified in higher termites, i.e. those termites that produce a proper royal jelly.
Your third comment:
Argument regarding reduction of expression of ß-1,4-endoglucanase relies on that expression decrease leads to a decrease in enzyme. Without measuring the activity of the enzyme in isolated glands, this is highly subjective. This maybe better as a suggestion for further directed work rather than support for the stated hypothesis.
There is very good correlation between expression as determined by qRT-PCR and enzyme activity in termite salivary gland as shown by a number of papers (e.g. references 11-19 in this manuscript).
Your fourth comment:
The “common sense” rule for statistics is not necessarily appropriate, as “common sense” is determined by the author to support their argument. Maybe best to present the data graphs and state that further work is necessary to corroborate the argument presented.
I removed the section on "common sense". It referred essentially to figure 5, where the expression of GHF9 clearly changes once the worker termites are isolated. Yet, although these numbers are quite convincing (as the number of spots in these SRAs for GHF9 are very large), one can not use statistics here.
I corrected all typos, but I did not italicize illumina, as this is never done in the litterature.
Reviewer 3 Report
The authors wrote a manuscript titled "The major protein in termite royal jelly" to analyze the differences of major proteins in termite royal jelly. Five termite genomes and two other cockroach species genomes were compared and public SRA data were mined for comparing gene expression. I can see the author performed a lot of work on analyzing these data but the topic of this manuscript is ambiguous. Although the author showed the difference of cellulose digesting enzymes, major allergens, and lysozymes between the termites of different castes, the results only described the number of genes, structural differences, and gene expression levels. If the author wanted to reveal the differences of protein components in the termites of different castes, the current analysis is preliminary and further experimental evidence of protein levels is lacking. More importantly, the current results couldn’t explain the contribution of these differences of proteins between the termites of different castes. The manuscript needed a great improvement to answer a specific scientific question.
Author Response
I thank you for the time and effort spent on reviewing my manuscript. Of the three reviewers your comments are the most critical, yet also not very specific. You write:
Although the author showed the difference of cellulose digesting enzymes, major allergens, and lysozymes between the termites of different castes, the results only described the number of genes, structural differences, and gene expression levels. If the author wanted to reveal the differences of protein components in the termites of different castes, the current analysis is preliminary and further experimental evidence of protein levels is lacking.
From this I understand that your major criticism is that the manuscript is lacking data on protein expression. I have never intended to analyze directly protein expression. I really wanted to identify nucleotide sequences of proteins expressed in salivary glands in order to be able to show whether or not expression of theses proteins might be correlated with the expression of one or more insulin-like peptides. This is why I start my introduction with insulin-like peptides that I previously identified. For some of these there is, admittedly far from perfect, evidence that they may stimulate protein secretion in specific circumstances. For example, IGF expression seems to be correlated with the start of physogastry in queens, while birpin expression is increased in those termites that destined to become either major workers and reproductives, while brovirpin expression seems to be associated with vitellogenesis (Veenstra, 2023). It seemed possible that one of the termite insulin/IGF-related peptides might specifically stimulate protein synthesis in the salivary gland, as explained in detail elsewhere (Veenstra, 2023). In order to do so, it is essential to know which proteins are expressed in the termite salivary gland and fortunately, a significant amount of work by others made it possible to address this question. Unfortunately, the current amount of SRA data is insufficient to answer the original question, i.e. is there a correlation between the expression of one of the IGF-related peptides and protein synthesis in the salivary gland. There are many different insulin/IGF-related peptides in insects and it is not always clear why there are so many of these hormones. Termites may be particularly well suited to address this question because of the existence of different castes on the one hand and the possibility/likelihood that, unlike in all other insect species studied so far, the different insulin-like peptides may be produced by different cell types. The latter possibility is only speculation for the time being, but since Periplaneta is very closely related to termites and in that species the different insulin-like peptides are produced by different cell types (Gen Comp Endocrinol. 2023 May 1;335:114233. doi: 10.1016/j.ygcen.2023.114233, freely available preprint at:
https://www.biorxiv.org/content/10.1101/2022.12.10.519892v2 ).
Round 2
Reviewer 3 Report
I have to say that I reread this paper more than ten times. I think the main problem with the paper is not only lacking the evidence on protein levels but also the topic of manuscript is confused. For example: The title was "The major protein in termite royal jelly." It could easily be misinterpreted that the manuscript mainly studied the protein components in termite saliva. However, when I read the full text, I found that the content of the study was on the genome level and transcriptome level. The more puzzling is the 2.4 section of the result. Most of SRA data for analyzing gene expression was not from the salivary glands, but from the whole body or head. Are the expression data comparable between different termites and different tissues? I could not believe that the data would reflect differences of relevant genes in the salivary glands of higher and lower termites, much less differences in protein composition. In the abstract, the authors say that the saliva components of higher termites are unknown, while the main saliva components of lower termites are cellulose digesting enzymes. A homolog of a cockroach allergen was found in both lower and higher termites, but it is in the latter that the salivary paralog gene got reamplified, facilitating an even higher expression of the allergen. However, the genome replication event couldn’t explain the composition of saliva. Just as cellulose digesting enzymes are not present in the saliva of higher termites, although it can be found in the genomes of both higher and lower termites. In addition, as shown in Result 2.1, the identification of GHF9 gene is not an intractable problem at present. If you really want to know the exact information about it, you only need to carry out the third-generation sequencing. Why bother with the result of assembling short sequences. And it's possible to draw the wrong conclusion. The author expressed concerns about the quality of the data he used in the results, such as the influence of the assembly quality of the genome on the number of gene copies identified and the influence of the number of transcriptome samples on the reliability of the conclusion. I applaud the authors for his realistic attitude but if the data are unreliable, how can we trust the conclusions drawn from them? More importantly, what is the specific contribution of these genes to the differentiation of higher termites and lower termites, and what are their effects? In summary, the authors may have originally intended to reveal differences between higher and lower termites by analyzing differences in the major proteins of their saliva, but the data could only reveal differences between genomes, not differences in tissue-specific expression levels, and eventually even deviated from the original question of the paper. Therefore, I would like to suggest that the author re-refines the topic of manuscript and improves the content of the manuscript before submitting it again.
Author Response
I thank you for time and effort on reviewing this manuscript. Just like you write you read the manuscript ten times, I also went over your comments and had problems understanding them, both in your initial review and now again. Nevertheless, if one reviewer does not understand the manuscript I must be doing something wrong, but I do not understand very well where or what. Below I try to explain my ideas, hopefully in a clearer fashion. I have added a paragraph and changed a few sentences that I think my clarify some of the problems, but I am not sure it will.
You start with writing: “I have to say that I reread this paper more than ten times. I think the main problem with the paper is not only lacking the evidence on protein levels but also the topic of manuscript is confused.”
I had no intention to look at protein levels directly in salivary glands as the first part of your comment seems to suggest. I got interested in looking at salivary gland proteins because I thought and I still think that some termite insulin-related peptides (which I prefer to call short IGF-related peptides as they evolved from IGF rather than insulin) may stimulate protein synthesis in salivary glands of workers. In order to be able to test this I needed to know which proteins are expressed in salivary glands. I think, but please correct me if I am wrong, that this is well explained in the introduction of the manuscript.
Others, whom I have cited (I hope I was complete, but if you think I have left someone out, please let me know, I will be happy to them to the list of references) have looked at the proteins produced in termite salivary glands. These proteins are mostly cellulose digesting enzymes in the lower termites, but, very interestingly, these same enzymes are no longer expressed by the salivary glands in higher termites. Data collected by Sillam-Dussès and colleagues in which they sequenced the major protein in salivary glands confirms this, a cellulose digesting enzyme in lower termites, but such proteins are lacking in higher termites. Instead, there is an ortholog of a major cockroach allergen in those species. Termite cockroach allergen orthologs are of two types (MA-1 and MA-2 in the manuscript). MA-1 lacks essential amino acids, which is probably not surprising given the difficulty that termites have to obtain sufficient amino acids. MA-2 on the other hand has these essential amino acids. Since in higher termites queens are fed exclusively with saliva, the cockroach allergen in saliva of higher termites must be of the MA-2 type. It is hard to imagine MA-2 being expressed outside the salivary glands, given the presence of the rare amino acid residues that it contains, while those amino acids are not needed for the biological activity that MA-1 must have (even though we do not really know this function). This would be waste of these amino acids that would be strongly selected against.
I thus feel that the expression of MA-2 in heads and thoraces can only be due to its expression in the salivary gland and this means that using its expression might be a way to establish a correlation with the expression of the various IGF-related peptides (it turns out the current data are insufficient to do so). Expression of MA-2 in higher termites may likely also be used as an indicator of whether the termite in question has a nursing or a building function. Some (very preliminary) data suggests it may be useful in that way. Furthermore, the amplification of the MA-2 gene in higher termites is consistent with the function of MA-2 as a protein source for queens and other colony members and this amplification mimics that of the major protein of honey bee royal jelly, even though these proteins have a completely different evolutionary origin.
In lower termites cellulose digesting enzymes are almost exclusively expressed in the salivary glands. Consequently, if one wants to look protein expression in salivary glands in lower termites, these proteins are ideal.
You write: “However, when I read the full text, I found that the content of the study was on the genome level and transcriptome level.”
I do not understand this comment. Proteins can be studied at the protein level, but also at the transcriptomic or genome level. So I do not understand what you seem to view as a contradiction.
You write: “Most of SRA data for analyzing gene expression was not from the salivary glands, but from the whole body or head. Are the expression data comparable between different termites and different tissues?”. I have tried to explain that above.
You write: “If you really want to know the exact information about it, you only need to carry out the third-generation sequencing.” I am not sure that I understand this. I point out that third generation sequencing is poorly equipped to get to the exact DNA sequences of highly amplified but poorly differentiated genes, such as the MA-2 genes in higher termites and the GHF9 genes in all termites.
Then you write: “More importantly, what is the specific contribution of these genes to the differentiation of higher termites and lower termites, and what are their effects?” Again I am not sure I understand this question. In a previous manuscript (Veenstra, 2023, PeerJ, full citation in the manuscript) I have attempted to use the published termite transcriptomes to see which of the various IGF-related peptides might play a role in the differentiation of the different termite castes. However, if you refer to the differentiation between higher and lower termites, it would seem that the expression of MA-2 makes it possible for worker termites to produce a true royal jelly, true in the sense that saliva from higher termite workers is a complete food source for survival. However, given the great variability in feeding patterns of higher termites I do not dare to suggest that this would be true for all higher termite species.